

# Linking muscle architecture and function *in vivo*: conceptual or methodological limitations?

Amelie Werkhausen[1], Øyvind Gløersen[2], Antoine Nordez[3,4], Gøran Paulsen[1], Jens Bojsen-Møller[5] and Olivier R. Seynnes[1]

[1] Department of Physical Performance, Norwegian School of Sport Sciences, Oslo, Norway
[2] Smart Sensors and Microsystems, SINTEF Digital, Oslo, Norway
[3] Movement - Interactions - Performance, MIP, Nantes Université, Nantes, France
[4] Institut Universitaire de France, IUF, France
[5] Department of Sports Science and Clinical Biomechanics, University of Southern Denmark, Odense, Denmark

Corresponding author
Amelie Werkhausen,
amelie.werkhausen@nih.no

## ABSTRACT

**Background**. Despite the clear theoretical link between sarcomere arrangement and force production, the relationship between muscle architecture and function remain ambiguous *in vivo*.

**Methods**. We used two frequently used ultrasound-based approaches to assess the relationships between vastus lateralis architecture parameters obtained in three common conditions of muscle lengths and contractile states, and the mechanical output of the muscle in twenty-one healthy subjects. The relationship between outcomes obtained in different conditions were also examined. Muscle architecture was analysed in panoramic ultrasound scans at rest with the knee fully extended and in regular scans at an angle close to maximum force (60°), at rest and under maximum contraction. Isokinetic and isometric strength tests were used to estimate muscle force production at various fascicle velocities.

**Results**. Measurements of fascicle length, pennation angle and thickness obtained under different experimental conditions correlated moderately with each other ($r = 0.40-.74$). Fascicle length measured at 60° at rest correlated with force during high-velocity knee extension ($r = 0.46$ at 400° s$^{-1}$) and joint work during isokinetic knee extension ($r = 0.44$ at 200° s$^{-1}$ and $r = 0.57$ at 100° s$^{-1}$). Muscle thickness was related to maximum force for all measurement methods ($r = 0.44-0.73$). However, we found no significant correlations between fascicle length or pennation angle and any measures of muscle force or work. Most correlations between architecture and force were stronger when architecture was measured at rest close to optimal length.

**Conclusion**. These findings reflect methodological limitations of current approaches to measure fascicle length and pennation angle *in vivo*. They also highlight the limited value of static architecture measurements when reported in isolation or without direct experimental context.

## INTRODUCTION

The structural arrangement of muscle fibres in skeletal muscles, also known as muscle architecture, is decisive for muscle function in animals and humans (*Gordon, Huxley & Julian, 1966*; *Lieber & Fridén, 2000*; *Narici, Franchi & Maganaris, 2016*). By *in vivo* measurements, the arrangement of muscle fibre bundles (fascicles) reflects the arrangement of sarcomeres, which in turn determines muscle fibre contraction properties, *i.e.,* functional excursion and maximal velocity and force. Thus, fascicle length is thought to reflect the number of sarcomeres arranged in series, and by proxy, the maximal contractile velocity and the capacity to produce force at high contractile velocity. In most human skeletal muscles, fibres act at an angle to the direction of the muscle force allowing a variety of fibre arrangements compared to fusiform muscles (*Gans & Bock, 1965*). For a given muscle volume, a greater fibre pennation would allow thicker fibres, with a greater number of myofibrils, or sarcomeres arranged in parallel, resulting in a higher force capacity (*Lieber & Fridén, 2000*).

Assessing muscle architecture *in vivo* has become relatively accessible in the past thirty years using ultrasound (see *Franchi et al., 2018* for review). Relatively low cost and quick implementation have led many laboratories to include the method routinely, to assess differences between populations or changes due to training (*e.g.,* *Blazevich et al., 2003*; *Kawakami et al., 1995*; *Stenroth et al., 2015*). Architecture outcome measures are often reported in isolation (*e.g.,* irrespective of muscle size), as a proxy to differences in muscle functional capacities (*e.g.,* *Lee et al., 2021*; *Ruiz-Cárdenas, Rodríguez-Juan & Ríos-Díaz, 2018*) raising two concerns. Firstly, the variation in methodological approaches and the important inter-rater dependence (*König et al., 2014*) of this technique have yielded discrepant findings and secondly, experimental evidence of the links between muscle architecture and muscle force production capacity, therefore, remains ambiguous.

Several studies have investigated the links between different measures of muscle size (*i.e.,* volume, physiological cross-sectional area) and force production capacity. Muscle volume and physiological cross-sectional area remain the main predictors for maximal force production (*Lieber & Fridén, 2000*; *Schantz et al., 1983*), but their composite nature conceals the specific influence of discrete architectural parameters (*i.e.,* fascicle length and orientation). A few studies have directly looked at the links between muscle architecture and force or joint torque production (*Ando et al., 2015*; *Trezise, Collier & Blazevich, 2016*; *Wakahara et al., 2013*). However, the results from these *in vivo* studies do not entirely agree with the theoretical links between sarcomere arrangement and function. For instance, a moderate relationship between maximum force and pennation angle was observed for knee extensor muscles (*Ando et al., 2015*) but not for the elbow extensor muscles (*Wakahara et al., 2013*). Similarly, indirect evidence of an interrelation between fascicle length and force production at high shortening velocities seems supported by some cross-sectional studies showing that sprinters had longer fascicle lengths than control subjects and distance runners (*Abe et al., 2001*; *Abe, Kumagai & Brechue, 2000*) but not by findings from a similar study (*Karamanidis et al., 2011*). By directly correlating muscle architecture measurements and torque, Maden-Wilkinson and colleagues observed that explosive torque production

(after 150 ms) was related to pennation angle but not to fascicle length, when estimated as the mean fascicle length from the four quadriceps muscles (*Maden-Wilkinson et al., 2021*). Conversely, when examining a single muscle (medial gastrocnemius), *Drazan, Hullfish & Baxter (2019)* found moderate correlations between gastrocnemius fascicle length and isokinetic torque and joint work. The relation between torque production capacity and the architecture of single muscles was also confirmed in the quadriceps muscles (*Trezise, Collier & Blazevich, 2016*), although not at velocities where the influence of fascicle length may be preponderant.

Inconsistent findings on the link between muscle architecture and function may stem from methodological issues. Different testing conditions including activation status, joint angle, and scanning technique (*Abe, Loenneke & Thiebaud, 2015*; *Franchi et al., 2018*; *Seynnes et al., 2009*) limit the comparison between existing studies. Architecture measurements are commonly performed at arbitrary joint angles at rest, although muscle architecture may better reflect force potential during maximal voluntary contraction (MVC), when residual slack is abolished, and the mean sarcomere length is a better predictor for force (*De Souza Leite & Rassier, 2020*; *Moo, Leonard & Herzog, 2017*). In addition, the estimation of fascicle length spanning beyond the field of view is dependent on calculation methods and may have the highest validity with panoramic scans (*Franchi et al., 2019*; *Scott et al., 2017*) or combined transducers (*Brennan et al., 2017*) to avoid errors due to extrapolation. Finally, the expression of muscle force measured in previous studies also differs (*e.g.*, joint torque, force contribution from synergistic muscles), tainting the relation to architectural parameters with the influence of other biomechanical factors.

This study aimed to take the above points into account to examine the relationships between muscle architecture and the mechanical output of the vastus lateralis muscle. Muscle architecture was captured during three experimental conditions based on different combinations of muscle contractile status and length: (i) at rest with the knee in anatomical position, at a knee angle close to optimal (ii) at rest and (iii) during MVC. Panoramic scans were performed for the condition at rest with the knee fully extended (as in several recent studies). Panoramic scans could not be performed for measurements during contraction. Therefore, we used synchronous recordings from two ultrasound transducers for the experimental conditions at optimal joint angle. Finally, we also scanned the fascicular behaviour under various contractile conditions and estimated the corresponding muscle force potential. We also scanned the fascicular behaviour under various contractile conditions and estimated the corresponding muscle force potential. Panoramic scans for muscle architecture analysis were performed at rest with the knee fully extended (as in most previous studies). Panoramic scans could not be performed for measurements during contraction, therefore we used synchronous recordings from two ultrasound transducers in these cases. We hypothesised that fascicle length would be positively correlated to force produced at high fascicle shortening velocity, and to the profile of the force-velocity relationship. We also hypothesised a positive correlation between fascicle length and muscle work. Furthermore, we hypothesised relationships between pennation angle and muscle thickness and maximum isometric forces. Lastly, we expected architecture measurements obtained under different experimental conditions (*i.e.,* joint configuration and contractile

status) to be moderately related, owing to inter-individual differences in muscle passive properties. For this reason, we also hypothesised that muscle architecture measured during maximum contraction at optimal joint angle would yield the strongest relationships to force production.

## METHOD

### Subjects and experimental protocol

Twenty-one healthy males ($n = 13$) and females ($n = 8$) (age: 27 $\pm$ 4 years; height: 176 $\pm$ 10 cm, body mass: 72 $\pm$ 11 kg) without any lower limb injury or neuromuscular disorder were included in the study. The ethical committee of the Norwegian School of Sport Sciences approved the study (approval number 95 – 090519), and all subjects gave written informed consent.

Participants performed a familiarisation session for the dynamometer measurements 1–3 days before data collection. On the day of data collection, ultrasound scans of resting muscle in the supine position were performed before the warm-up. After a standardized warm-up of 10-min cycling, the subjects performed maximum voluntary contractions during isometric and isokinetic tests for the knee extensor muscles. All tests were performed on the same, randomly selected leg.

### Knee extension strength tests

Participants were fastened in an isokinetic dynamometer (IsoMed2000; D&R Ferstl, Hemau, Germany) with a hip angle of 80°. The rotation axes of the knee and dynamometer arm were aligned, and the lower leg was fastened to the customised dynamometer arm. To avoid artefacts due to foot pad compliance, we replaced the compliant pad with a shin protector, attached to the dynamometer *via* a 3D-printed adapter to increase the stiffness of the interfaced pad compared to the original one. After a specific warm-up consisting of five contractions at 50% and 80% effort, participants performed two sets of maximum isometric contractions at six different angles (100°–50° knee flexion, in steps of 10°) in randomised order. They were asked to increase the torque over 3 s and hold it for 3 s. Subsequently, two sets of isokinetic knee extensions (range of motion from 110° knee flexion to 0°) at five different velocities (50, 100, 200, 300, and 400° s$^{-1}$) were performed in randomised order. The best of the two attempts (*i.e.,* with the highest torque) at each angle and velocity was used for all further analyses. In addition, we recorded the torque during a passive movement through the whole range of motion for gravity and passive torque correction. To minimise the effects of fatigue, participants rested for three minutes between each effort.

Knee extensor torque was filtered using a second-order bi-directional Butterworth filter with a cut-off frequency of 20 Hz and corrected by subtracting the torque measured during the passive movement. Subsequently, knee extension force was calculated as the quotient of the knee joint torque and the knee joint angle-dependent moment arm, following the method suggested by *Bakenecker, Raiteri & Hahn (2019)*. Vastus lateralis force was estimated as the relative contribution of the muscle to knee extension force based on its physiological cross-sectional area obtained in previous studies (*Akima et al., 1995*). Joint-

and muscle work were calculated over the iso-velocity phase common to all isokinetic tests (80−50°) by trapezoidal numerical integration using joint torque and angle, and muscle force and muscle length. Muscle length changes were estimated using the product of fascicle length and the cosine of the pennation angle.

## Assessment of muscle architecture

Panoramic scans ($REST_0$) of the vastus lateralis muscle were taken while subjects lay fully relaxed in a supine position (knee angle at 0°) with the feet immobilized in a standardized position, securing no rotation of the hip, using a 50 mm ultrasound transducer (L 12-5 (transducer) and HD11XE Philips Ultrasound; Philips, Andover, MA, USA). A scanning path was drawn on the skin at 50% of the distance between the greater trochanter and the distal end of the muscle, over the thickest portion on the mediolateral axis of the muscle. The transducer was moved along the scanning path (*i.e.,* along the length of the muscle) proximally during acquisition, until a region large enough to include one whole fascicle length was scanned. Participants were reminded to relax their muscles during the scan. We analysed the panoramic scans using a modified version of an open-source plugin (*Seynnes & Cronin, 2020*) for ImageJ (National Institutes of Health, Bethesda, MA, USA), to automatically detect dominant fascicle orientations in superficial and deep regions and reconstruct a composite fascicle with a simple spline fitting (Fig. 1). A region of interest was selected in the images to account for small differences in the dimension of panoramic scans. Muscle thickness was computed as the mean distance between the two aponeuroses, pennation angle as the angle between the deep aponeurosis and fascicle orientations, and fascicle length as the length of the curve fitted to the deep and superficial fascicle fragments, between aponeuroses.

Since panoramic scans cannot be performed during muscular contractions, we developed a 3D-printed structure to hold two flat-shaped ultrasound transducers (LV7.5/60/96Z LS128 (proximal) and LV8-5N60-A2 ArtUs (distal), Telemed, Vilnius, Lithuania) in series. The holder differed from similar devices described by other groups (*e.g., Brennan et al., 2017*) by being angled between transducers. Measurements of architecture being sensitive to the pressure exerted on muscle tissue, we sought to limit a heterogenous pressure distribution that a straight 20 cm device could cause when pressed on the curved shape of the thigh. Pilot tests indicated that a 5° angulation in the longitudinal axis of the transducers allowed consistent contact with the skin along the length of transducers, with limited and relatively homogeneous pressure along the holder, as evidenced by ultrasonographic scans. This angle was used for all participants. The probes were placed along the direction of the muscle fibres (in the muscle frontal plane) with the proximal end of the distal probe at 50% of the muscle length and were fastened using self-adhesive tape and elastic bands (Fig. 2). Ultrasound scans were taken during isometric and isokinetic tests to build individual force-length and force-velocity curves (see next section). Ultrasound data were synchronised by recording a digital trigger signal from both machines. Scans from ultrasound frames preceding isometric contractions at a knee angle of 60were also analysed and labelled $REST_{60}$.

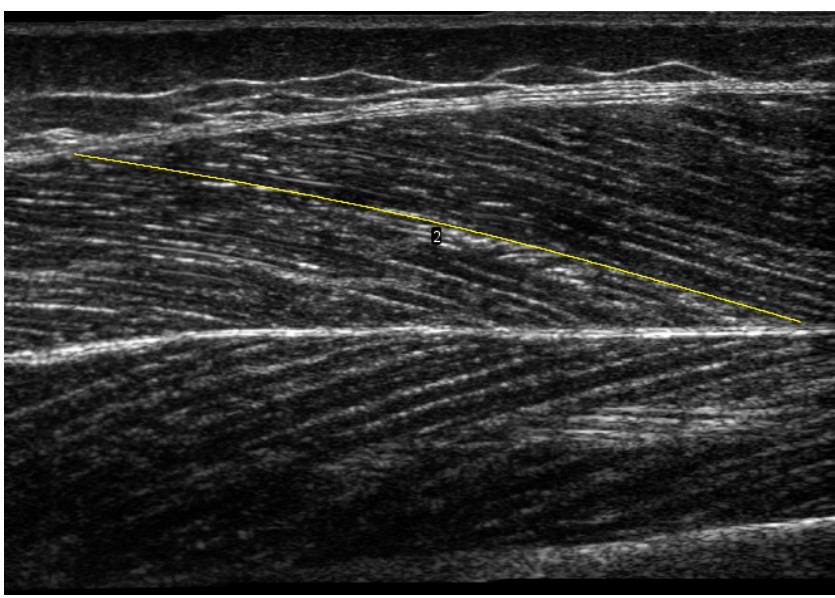

**Figure 1   Panoramic ultrasound scan of vastus lateralis muscle.** The yellow line illustrates a composite fascicle, obtained from all fascicle fragments detected by the automatic analysis.

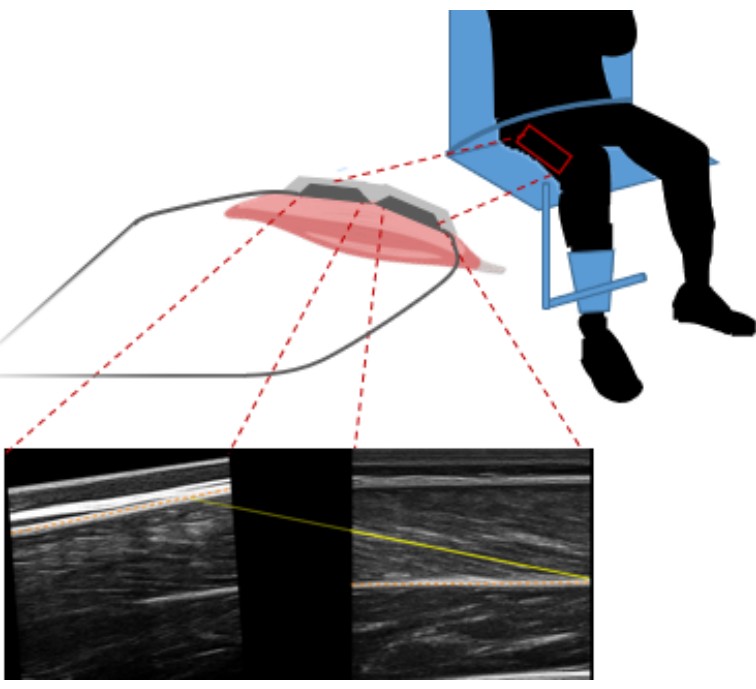

**Figure 2   Ultrasound setup during the knee extension strength test with example ultrasound scans of vastus lateralis muscle from the proximal (left) and distal (right) ultrasound transducer.** The orange dashed lines represent the aponeuroses, the yellow solid yellow line represents the fascicle segmentation, and the yellow dotted line represents the extrapolated part of the fascicle.

Scans were analysed by manual segmentation of the two images individually using UltraTrack software (*Cronin et al., 2011*; *Farris & Lichtwark, 2016*). The coordinates of aponeuroses and fascicles obtained from segmentation were combined by setting them in a common coordinate system, using linear and rotational transformations (*i.e.,* to account for the relative position and orientation of the probes). For the estimation of fascicle length, we had planned to account for fascicle curvature by using the deeper portion of the fascicle in the distal scans and the superficial portion in the proximal ones. Unfortunately, the quality of most proximal scans was insufficient to segment fascicle orientation reliably. Proximal scans were therefore used to determine the orientation of the upper aponeurosis only. The line representing the fascicle started in the lower right corner of the image from the distal transducer and was linearly extrapolated to the upper aponeurosis (Fig. 2). During contraction (*i.e.,* fascicle shortening and pennation angle increase), fascicles extrapolated out of the distal field of view did not always span to the proximal one, because of the gap between the transducers' arrays. In such cases, the superficial insertion of the fascicles was set to the linear extrapolation of the superficial aponeurosis from either field of view, whichever yielded the shortest fascicle length.

## Muscle contractile properties

Fascicle force-length data were fitted using MATLAB (MathWorks, Natick, MA, USA) with a physiologically appropriate model (*Azizi & Roberts, 2010*, Eq. (2); *Brennan et al., 2018*, Eq. (1)) to determine the theoretical maximum force:

$$F = force_{\max} \cdot e^{-\left| \left( \frac{\left( \frac{L}{L_0} \right)^b - 1}{s} \right) \right|_2} \tag{1}$$

where $F$ is fascicle force, $force_{\max}$ is the theoretical maximum force, $L$ is fascicle length, $L_0$ is fascicle length at $force_{\max}$, $b$ is skewness, and $s$ is the width of the curve. The $force_{max}$ parameter is therefore determined by the accuracy of the fitted function and may differ from measured maximum force (*e.g.*, this is the case for the mean $force_{max}$ and the mean – measured – peak force).

During the isokinetic trials, muscle architecture was assessed throughout the range of motion using a semi-automated tracking algorithm (*Cronin et al., 2011*; *Farris & Lichtwark, 2016*). We exported the coordinates of the labelled images from the software and calculated fascicle length similarly to the condition with fixed joint angle. Fascicle velocity was calculated using the central difference method. Subsequently, peak fascicle velocities and peak fascicle forces were measured during the dynamometer iso-velocity phase of each isokinetic condition. Collected velocities and forces were then combined with the theoretical isometric maximum force ($force_{\max}$, Eq. (1)) to produce individual force-velocity curves:

$$F = force_{\max} \frac{\left( 1 - \frac{v}{v_{max}} \right)}{1 + G * v / v_{max}} \tag{2}$$

Where $v$ is fascicle shortening velocity, $v_{max}$ is the maximum shortening velocity, $G$ is the curvature parameter and $F$ is the force capacity (at the corresponding velocity). Similar

**Table 1  Measurements of vastus lateralis fascicle length, pennation angle and thickness.**
Measurements were performed at rest with fully extended knee ($REST_0$), at rest with the knee angle of 60° ($REST_{60}$) and during MVC with the knee angle of 60° ($MVC_{60}$). Data are mean $\pm$ standard deviation.

|  | Length$_{fascicle}$ | Pennation angle | Thickness |
|---|---|---|---|
| $REST_0$ | $82.3 \pm 9.5$ | $15.7 \pm 2.4$ | $21.2 \pm 2.7$ |
| $REST_{60}$ | $120.4 \pm 12.7$ | $10.9 \pm 1.6$ | $21.3 \pm 2.3$ |
| $MVC_{60}$ | $96.7 \pm 18.1$ | $14.5 \pm 2.7$ | $23.7 \pm 3.0$ |

to *Brennan et al. (2018)*, we allowed a curvature of $3 < G < 9$. We did not constrain $v_{max}$ because we measured fascicle shortening velocity at different isokinetic velocities.

We used the force-length and force-velocity relationships to estimate three measures of force of the knee extensor muscles: maximum force ($force_{max}$), the force produced during the fastest ($400°\ s^{-1}$) isokinetic condition ($force_{iso400}$), the curvature parameter $G$, and the slope between the force produced at the fascicle velocity measured during the slowest ($50°\ s^{-1}$) and fastest isokinetic ($400°\ s^{-1}$) velocities, normalised to $force_{max}$ ($force_{slope}$). The parameter $force_{slope}$, therefore, represents the loss in force due to an increase in fascicle shortening velocity. It was introduced as a simplified measure of the curvature of the force-velocity relation ($G$-parameter), for which the influence of noise from the high number of differently measured parameters and calculations may limit the sensitivity of correlation analyses. The four force parameters were correlated with the architectural parameters.

### Statistical analyses

All data were tested for normal distribution using the Kolmogorov–Smirnov test. As a first step, we tested correlations between muscle architecture measurements obtained with different methods. The Pearson product-moment correlation matrix was calculated for measurements of fascicle length, pennation angle and thickness measured at $REST_0$, $REST_{60}$ and $MVC_{60}$. We also assessed the relationships between the muscle architecture parameters measured in the three different conditions ($REST_0$, $REST_{60}$ and $MVC_{60}$) and estimated from the force-length curves, and the force parameters ($force_{max}$, $force_{iso400}$, $G$ and $force_{slope}$) using Pearson correlation. Correlation coefficients were considered strong for $r \geq 0.7$, moderate for $0.7 < r \geq 0.4$, and weak for $r < 0.4$ (*Schober, Boer & Schwarte, 2018*). Significance was defined at $\alpha = 0.05$ and all tests were conducted using MATLAB.

### RESULTS

### Muscle structure and force measurements

Vastus lateralis muscle architecture measurements differed as expected between the three different experimental conditions (Table 1). Correlations between the different measurements of muscle thickness, fascicle length and pennation angle, respectively, were moderate to strong comparing most conditions, although we did not find statistically significant correlations between fascicle length measured at $REST_0$ and $REST_{60}$, and between $REST_{60}$ and $MVC_{60}$ (Fig. 3).

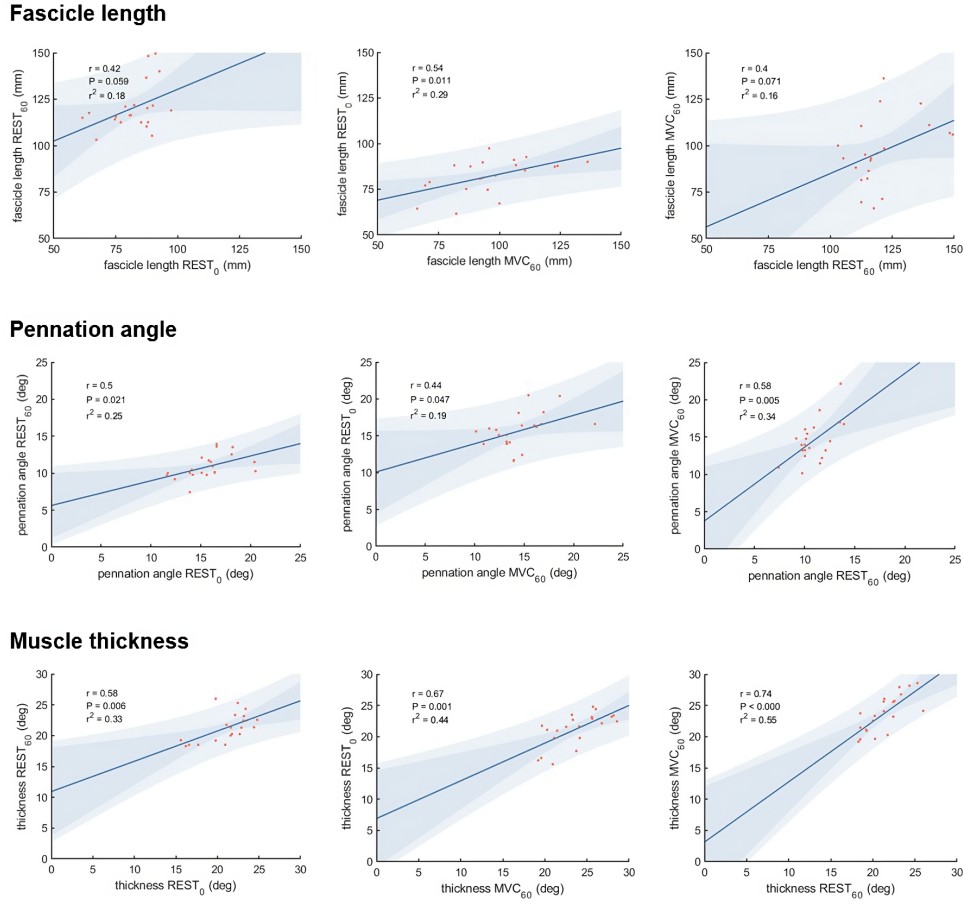

**Figure 3** **Scatter plots with linear regression lines for muscle architecture measurements in different conditions:** At rest with the knee fully extended ($REST_0$), at rest with the knee in 60° flexion ($REST_{60}$) and during a maximum contraction ($MVC_{60}$). The shaded area shows the 95% observational and functional confidence intervals ($n = 21$, DOF = 19).

Optimal fascicle length during maximum contraction, calculated using the muscle force-length curve, was on average $105 \pm 17$ mm. The curvature of the force-velocity curve, described by the parameter $G$, was $6.1 \pm 2.7$ and the estimated maximum fascicle shortening velocity ($17.5 \pm 10.5\ L_0\ s^{-1}$), where $L_0$ is the optimal fascicle length. force$_{slope}$ (as calculated between the normalised theoretical force measured at the slowest and fastest condition) was $-0.22 \pm 0.13$ N⋆cm$^{-1}$⋆s. Force-length and force-velocity curves are shown in Fig. 4. Joint work ranged from $43 \pm 14$ J to $76 \pm 18$ J and muscle work ranged from $15 \pm 13$ J to $44 \pm 29$ J over the five different isokinetic velocities.

The theoretical maximum force of the knee extensor muscles was $5295 \pm 1504$ N and the knee extension force produced at the highest fascicle velocity was $2054 \pm 592$ N. The estimated maximum vastus lateralis fascicle force was $1856 \pm 536$ N.

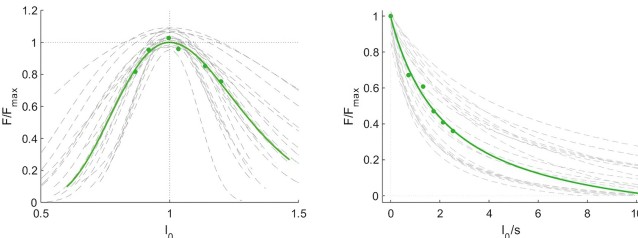

**Figure 4** **Force-length and force-velocity properties of the vastus lateralis muscle from individual participants (grey) and group mean data (green).** Force data were normalised to the maximum force (Force/Force_max), fascicle length to the optimal length ($l_0$) and fascicle velocity is expressed as fascicle lengths per second ($n = 21$).

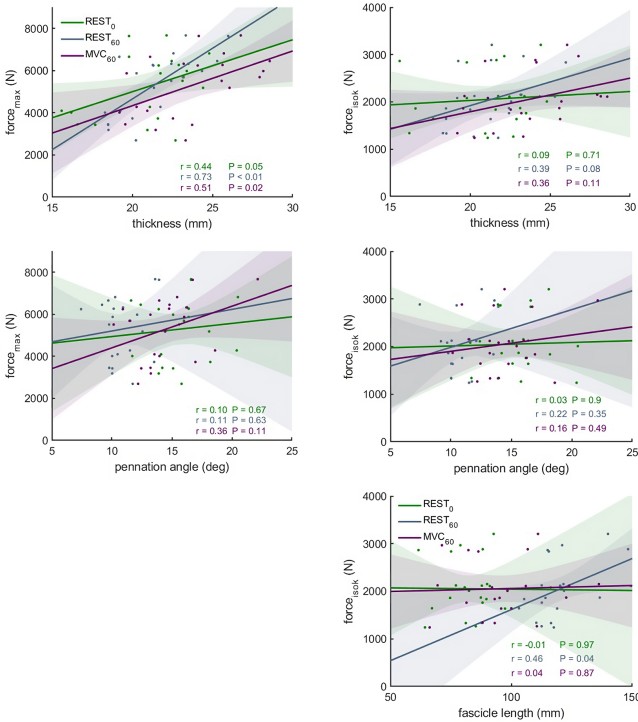

**Figure 5** **Scatter plots with linear regression lines (±95% confidence interval) and Pearson correlation coefficients between force and muscle architecture measurements.** Relationships between maximum force (force_max) and the force produced during the fastest velocity in the isokinetic test (force_isok) and muscle architecture measurements in different conditions: At rest with the knee fully extended (REST_0), at rest with the knee in 60° flexion (REST_60) and during a maximum contraction with the knee in 60° flexion (MVC_60) ($n = 21$, DOF = 19)).

## Relationships between muscle structure and force

The correlations between muscle architecture parameters and force measured in the three experimental conditions varied from negligible to strong. Correlation coefficients for force_max and force_iso400 and muscle architecture are shown in Fig. 5. Force produced at the highest velocity was moderately correlated with fascicle length measured at REST_60,

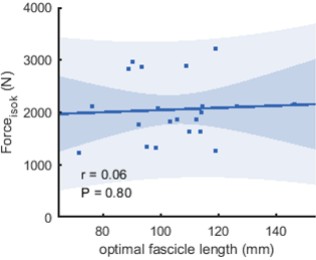

**Figure 6  Scatter plots with linear regression lines (±95% confidence interval) and Pearson correlation coefficients between force and optimal fascicle length.** Force is produced during the fastest velocity in the isokinetic test (force$_{isok}$) and optimal fascicle length was calculated from the force-length relationship ($n = 21$, DOF = 19)).

**Table 2  Pearson correlation coefficients for fascicle length measured in different conditions and during a maximum contraction.** Measurements were performed at rest with the knee fully extended (REST$_0$), with the knee at rest in 60° flexion (REST60) and during a maximum contraction with the knee in 60° flexion (MVC60); knee joint work and vastus lateralis fascicle work were measured at different isokinetic velocities. $*P < 0.05$, $**P < 0.01$, $n = 21$, DOF = 19.

| | | REST$_0$ | REST$_{60}$ | MVC$_{60}$ |
|---|---|---|---|---|
| Joint work | 400° s$^{-1}$ | 0.22 | 0.38 | −0.02 |
| | 300° s$^{-1}$ | 0.22 | 0.38 | 0.01 |
| | 200° s$^{-1}$ | 0.21 | 0.44* | −0.14 |
| | 100° s$^{-1}$ | 0.29 | 0.55** | −0.05 |
| | 50° s$^{-1}$ | 0.05 | 0.33 | −0.27 |
| Muscle work | 400° s$^{-1}$ | 0.11 | 0.18 | −0.14 |
| | 300° s$^{-1}$ | −0.12 | 0.15 | −0.20 |
| | 200° s$^{-1}$ | −0.03 | 0.13 | −0.18 |
| | 100° s$^{-1}$ | 0.25 | 0.13 | −0.07 |
| | 50° s$^{-1}$ | 0.06 | 0.23 | 0.13 |

whereas there were no significant relationships with fascicle length measured in the other two conditions. Likewise, fascicle length at theoretical peak force (estimated from the force-length relationship) was not related to force produced at the highest velocity (Fig. 6). Pennation angle measured in all conditions did not correlate with either force parameter. However, maximum force was strongly correlated with muscle thickness measured at REST$_{60}$ and moderately with thickness measured at REST$_0$ and MVC$_{60}$. Force produced at the highest velocity was not correlated with muscle thickness in either condition. The calculated slope parameter and the force-velocity curvature G had no significant correlations with any of the measured muscle architecture parameters for all three experimental conditions (all $r < 0.24$ and $P > 0.30$ for slope and r $<.15$, $P > 0.51$ for G). Fascicle length had no significant correlations with joint or muscle work when measured at REST$_0$ and MVC$_{60}$ but correlated with joint work at 100 and 200 ° s$^{-1}$ when measured at REST$_{60}$ (Fig. 5, Table 2).

## DISCUSSION

This study assessed the relationships between vastus lateralis muscle architecture measurements and muscle force production. Against our hypothesis, we did not find significant correlations between fascicle length or pennation angle-measured at $MVC_{60}$, $REST_0$ or estimated at theoretical peak force—and any measure of force production. Muscle thickness was, however, related to maximum force independently of the measurement method. In line with our second hypothesis, most correlation coefficients were higher for $MVC_{60}$ than $REST_0$, but, surprisingly, measurements from $REST_{60}$ showed the highest relationship to force parameters.

### Relationship between muscle architecture and force

Using more complex models of muscle performance (*i.e.,* sprint time), certain studies suggested a clearer functional relevance of fascicle length for rapid force production (*Abe, Kumagai & Brechue, 2000*; *Kumagai et al., 2000*). Likewise, moderate correlations were reported between pennation angle and isometric torque (*Wakahara et al., 2013*, $r = .47$). These results are not in accordance with those of the present study. The lack of relationship between variables capturing fascicle conformation and force production may be partly attributable to a higher relative contribution of other intrinsic muscle properties, such as fibre type composition, muscle volume, muscle site variations, muscle activation deficits, and external factors such as moment arm (*Lee & Piazza, 2009*; *Williams et al., 2008*). However, considering recent literature, we also attribute the lack of correlations (supporting the theoretical links) between fascicle length and pennation angle with $force_{iso400}$ and $force_{max}$ to inaccurate representations of sarcomeres arrangement with the current methods (*Lieber, 2022*). The assessment of 3D structures in 2D may be another factor contributing to the inaccurate representation (*Rana, Hamarneh & Wakeling, 2014*). In animal studies, where it is common to measure sarcomere length, several studies have shown a connection between optimal fascicle length and sarcomere length (*e.g.*, *Bodine et al., 1982*; *Salzano et al., 2018*).

Recent *in vivo* measurements using microendoscopy showed that, if fascicle length changes were reflected in sarcomere length, the strength of this correlation was only moderate in different muscles (*Lichtwark et al., 2018*; *Pincheira et al., 2021*). The same authors also pointed out that within-muscle regional differences in muscle architecture may confound the association between in-series sarcomere number and fascicle length. We attempted to mitigate confounding factors by implementing several fascicle length measurements and extensive tests of muscle force capacity during rapid movements. Our comparisons of vastus lateralis muscle architecture measurements with the curvature parameter $G$ and $force_{slope}$ aimed to expand the representation of muscle force-generating properties. The absence of significant correlations of $G$ and $force_{slope}$ with fascicle length could be due to the limited validity of fascicle length measurements to assess contractile properties. Of note in the case of $G$, the curvature of the force-velocity curve, we cannot exclude errors affecting the model of the force-velocity relationship. This variable is not commonly reported in human studies. *Brennan et al. (2018)* reported higher values ($G = 8.9 \pm 0.5$) for the vastus lateralis fascicles, which indicates greater curvature (with

lower std), compared to our findings ($G = 6.1 \pm 2.7$). The reasons for this discrepancy are not clear but $G$ or the more direct slope of force decline used here can theoretically capture the muscle's capacity to retain force when movement velocity increases. The lack of correlation between these variables and fascicle length may be more attributable to the limitations of the current *in vivo* approaches than to scaling differences between models of the force-velocity relation. Such methodological limitations may on the other hand not affect joint work, a macroscopic expression of muscle energy production requiring fewer assumptions. We found moderate relationships between fascicle length, measured at $REST_{60}$, and joint work at 100 and 200 $°\ s^{-1}$. These correlations are in accordance with findings from *Drazan, Hullfish & Baxter (2019)* and indicate that longer fascicles enable greater work at the joint at these velocities. Yet this interpretation was not supported by correlations between fascicle length and muscle work. Overall, the weak correlation between fascicle length and expressions of muscle mechanical output is in part attributable to assumptions and methodological factors affecting muscle architecture measurements and force estimations, as discussed in the section on *methodological considerations*.

The relationship between the number of in-parallel sarcomeres and pennation angle is even less documented. The only study investigating this relation indirectly to date did not evidence any correlation between mean fibre cross-sectional area (a proxy to the number of in-parallel sarcomeres) and pennation angle (*Henriksson-Larsen et al., 1992*). Although ultrasonographic measurements of architecture reflect the number and organisation of sarcomeres to an extent, the level of validity of this relationship remains unknown. These results support the view that the significance of pennation angle for force production may be smaller than previously thought (*Lieber, 2022*).

Contrary to fascicle length and pennation angle measurements, moderate to strong correlations were found between muscle thickness and $force_{max}$, although this relationship was–expectedly–lost when looking at explosive force ($force_{iso400}$). Studies looking at the association between this relatively simple measure of muscle size and strength are surprisingly scarce. Previous reports however suggest a moderate correlation between thickness of the whole quadriceps (*Freilich, Kirsner & Byrne, 1995*), and discrepant results regarding single muscles. For example, Ando and colleagues found the thickness of the vastus intermedius to be strongly correlated to maximal strength ($r = .74$), while no correlation was observed for the vastus lateralis (*Ando et al., 2015*). The previous and present findings suggest that muscle thickness may provide a more comprehensive representation of maximal force potential than pennation angle, albeit with a moderate accuracy. Of course, force potential is better reflected by more exhaustive measures of muscle contractile tissue, such as anatomical- or physiological cross-sectional area. Our intention here was merely to investigate the extent to which discrete architecture parameters are representative of force production capacity. Taken together, these findings highlight their limited significance in this context, when reported in isolation.

The differing results of for example the study by *Ando et al. (2015)* could also be due to physiological and mechanical differences between the quadriceps muscle heads. As in our study, the vastus lateralis is often used as representative for all heads due to its accessibility.

## Comparison of muscle architecture measurements obtained in different conditions

What difference does the choice of methodological approach make on the estimation of force potential from architecture? Muscle architecture methods used in this study (panoramic scan at rest and composite scan during MVC) were selected to represent the current perception of theoretical optimality, within technical boundaries (*i.e.,* no panoramic scan during contraction). They presented unexpectedly low correlations between each other for fascicle geometry ($r = .40-.54$ for fascicle length measurements and $r = .44-.58$ for pennation angle measurements), while moderate to strong correlations were observed for muscle thickness ($r = .58-.74$). Despite the inter-individual variability in architectural changes due to contracting state and joint configurations, we expected these common methods to yield more similar results. This discrepancy is likely due to physiological and methodological causes (scanning technique and image analysis). Amongst physiological factors, the relatively moderate agreement between measurements of architecture in different conditions could be related to the inter-individual variability in architectural changes. Changes in muscle length and activation may affect fibre length inconsistently across participants, as suggested by the nonuniformity of sarcomere length change during *in situ* contractions (*Julian & Morgan, 1979*).

The expected robustness of the data collected in each condition relies on different parameters. Despite our precautions to reduce variability with a single investigator recording ultrasound data (AW) and the use of semi-automated software to analyse architecture, the different scanning approaches (panoramic *vs.* combined field of views) likely clouded the present comparisons. Panoramic scanning was not technically possible for $MVC_{60}$, despite the advantage of this modality to abolish sarcomere nonuniformity at rest and during force development and arguably providing more valid estimates of fascicle length (*e.g., De Souza Leite & Rassier, 2020*). If the $REST_{60}$ method has been used in the past (*e.g., Seynnes, De Boer & Narici, 2007*), it was the least expected to yield relations to force production. Panoramic scanning was not used for the condition $REST_{60}$, as scans were taken before the start of the $MVC_{60}$ condition (same angle, with MVC contraction), with the same system of ultrasound probes fastened over the muscle. Yet, $REST_{60}$ was the only condition where fascicle length was correlated with $force_{iso400}$ and with joint work. This relationship, lacking with $REST_0$, could indicate a large influence of inter-individual variation in tissue morphology and material properties, *e.g.,* muscle slack length, which has been suggested as an additional source of noise (*Aeles et al., 2017*). With the $REST_{60}$ condition, the advantage of measuring fascicle length near optimal angle could also outweigh the errors due to simplification or extrapolation of fascicle geometry. In turn, the missing correlation for $MVC_{60}$ could indicate that factors that were unaccounted for, such as increased curvature or pressure due to bulging, outweigh the advantage of standardising fibre and sarcomere length. The angled double probe setup was chosen to minimise the effects of pressure caused by long and flat probes. Whilst it arguably allowed a more even pressure distribution along the scanning area, it might not have mitigated the artefact caused by compression sufficiently. Future studies may improve measurements from

double probes by adopting a design limiting external pressure and by ensuring sufficient image quality to compute fascicle curvature.

Overall, the unexpectedly low agreement between the different conditions of measurements confirms the importance of choosing appropriate methods to evaluate muscle architecture. The present data do not allow distinguishing between the effects of scanning technique, muscle configuration and image analysis method. They suggest, however, that scanning at a joint angle close to optimal is better than an arbitrary joint configuration.

### Methodological considerations

As is typically the case with *in vivo* studies, the present results are limited by necessary assumptions. These were required here, to obtain comprehensive force estimates using input data from isometric and isokinetic tests and a model of force-length and force-velocity properties similar to *Brennan et al. (2018)*. Modelling of muscle function relies on best-fit approximations and choices of normative variables. There are several possible approaches to estimating force and velocity values. Because fascicle lengths are shorter at greater isokinetic velocities (due to elastic tissue stretch), measurements could not be taken at optimal fascicle length. A previous study had therefore averaged force and velocity measurements over a dynamometer range with constant velocities (*Hauraix et al., 2017*). Here we used a similar approach using values occurring in the same range (50–80)°. However, we retained the peak values from this phase to avoid overestimating the force decline when constructing the force-velocity relation. In addition, we estimated the joint angle-dependent moment arms as suggested in the literature (*Bakenecker, Raiteri & Hahn, 2019*) because of the limitations inherent to anthropometric measures (*Tsaopoulos, Maganaris & Baltzopoulos, 2007*). It is, however, possible that individualised measures of moment arm based on MRI scans of the joint could have improved our estimates of force. Our force models were also likely affected by the fact that the non-planar dynamics of fascicles (*e.g.*, *Rana, Hamarneh & Wakeling, 2014*) and history dependence of force and fascicle length (*e.g.*, *Stubbs et al., 2018*) were not taken into account. Further studies including conditioning contractions should ascertain that the 3-minute intervals between measurements and the random order of the tests were sufficient to mitigate the effect of history dependence.

In addition, estimating muscle force relies on necessary assumptions *in vivo*. Our estimates were based on relative physiological cross-section area and moment arms from the literature because MRI measurements were not available. We cannot discard the potential influence of inter-individual differences in the relative size, moment arms (*Trezise, Collier & Blazevich, 2016*) and activity of synergistic muscles. In consideration of the methodological limitations to estimating muscle force, we chose to refrain from calculating fascicle force, albeit a more direct expression of the force production related to fascicle length. This decision was further based on the uncertainty of the current models of force transfer from fascicles to the tendon (see the discussion on this issue in *Lieber, 2022*). While the approaches taken in this study seem reasonable and in phase with recent literature, the accuracy of our estimates of force was unavoidably affected by methodological choices.

## CONCLUSIONS

Apart from muscle thickness, which can be seen as a compound representation of sarcomere number in parallel, we did not consistently find relationships between fascicle conformation and force production. Despite some moderate correlations between fascicle length and joint work under certain velocities, neither of the two methods currently perceived as best ($REST_0$ and $MVC_{60}$) correlated to muscle force or work production. The only statistically significant link between fascicle variables and force was found when the muscle was scanned at rest, with the joint angle close to optimal. Static measurements of muscle architecture (at rest or during isometric contraction) have an undeniable physiological relevance. However, these findings suggest that they should be collected with the muscle close to optimal length, and that caution should be exerted in their interpretation as a predictor for muscle performance when other relevant anatomical variables are unknown.

### Funding

The authors received no funding for this work.

### Competing Interests

The authors declare there are no competing interests.

### Author Contributions

- Amelie Werkhausen conceived and designed the experiments, performed the experiments, analyzed the data, prepared figures and/or tables, authored or reviewed drafts of the article, and approved the final draft.
- Øyvind Gløersen conceived and designed the experiments, prepared figures and/or tables, authored or reviewed drafts of the article, and approved the final draft.
- Antoine Nordez, Gøran Paulsen and Jens Bojsen-Møller conceived and designed the experiments, authored or reviewed drafts of the article, and approved the final draft.
- Olivier R Seynnes conceived and designed the experiments, analyzed the data, authored or reviewed drafts of the article, and approved the final draft.

### Human Ethics

The following information was supplied relating to ethical approvals (i.e., approving body and any reference numbers):

The Ethics committee of the Norwegian School of Sport Sciences granted ethical approval to carry out the study (Ref: 95 -090519).

### Data Availability

The data is available in the Supplementary File.

## Supplemental Information

Supplemental information for this article can be found online at http://dx.doi.org/10.7717/peerj.15194#supplemental-information.

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
