# Peer review of "Linking muscle architecture and function in vivo: conceptual or methodological limitations?"

_PeerJ, doi:10.7717/peerj.15194_

## Round 0.1 · original submission · Major Revisions

The reviewers do see merit in your work but they have pointed out substantial limitations in the methodology applied (.e.g focus on VL muscle architecture only, measuring thickness instead of PCSA, lack of data on muscle activation, etc. I would invite you to either address the limitations identified or carefully consider these (the remaining limitations) in the interpretation of the results.

·

Basic reporting

In general, the goal of this work seems to confirm that certain architectural parameters are linked to muscle mechanical properties. However, the background for this often provided by animal studies is not included. If the methodological limitations are really responsible for the weak correlations, then very strong correlations should have been found in animal studies. A discussion of the animal literature is this regard is missing.

The authors talk about using two different ‘measurement approaches’, ‘measurement methods’ or ‘US-based approaches’, but they compare US images obtained for different joint angles and different contraction states. The reader is misled, for example in the abstract, by the description that different approaches are compared suggesting a comparison between panoramic US imaging and US scanning with two in series probes. Actually, two different approaches were used to obtain muscle architectural parameters for three different muscle conditions. This needs to be adjusted throughout the manuscript.

See additional comments for specific comments.

Experimental design

This manuscript describes a study in which architectural parameters of M. vastus lateralis were correlated to various parameters representing the mechanical muscle properties. This is not new, but the authors argue that the methods used in previous studies had important limitations, leading to ambiguous results. Although some limitations were considered in this study, many limitations remain and, hence, similar results as reported previously were found. To really make a step forward in understanding these relationships, a more thorough approach is needed. This does not require the development of new methods, because they have all been applied before.
- The authors measure muscle thickness instead of the CSA or PCSA. The latter two are a better representation of the amount of contractile material in parallel and can be assessed using US. Because this was not assessed, literature was used to distribute the force among the quadriceps muscle (see ln 160). Using individual assessment of PCSA would provide a much more accurate assessment of VL force.
- The authors assume that fascicle length reflects the number of sarcomeres in series (e.g. ln 48), while with the same number of sarcomeres in series many different fascicle lengths can be obtained. Without knowing sarcomere length, the number of sarcomeres in series cannot be assessed using US. However, an estimate may be obtained by comparing the fascicle length at optimal force. As the length-force characteristics were estimated by the authors, this more meaningful length could be used.
- The authors raise a concern about inter-rater dependence of assessing muscle architecture using US. However, their study does not report on this. Also, it is not indicated anymore in the discussion. Does it mean that with the applied methods, inter-rater variability is not an issue anymore?
- It has been proposed that properties of passive muscles are dependent on history effects (see papers Stubbs and Herbert). To attenuate such effects, conditioning contractions should be performed. Especially when different joint angles, including those for which slack lengths are expected, are tested. This reduces variability and is thus like to affect the relationships between architecture and mechanics.
- Assessment of mechanical properties of passive muscles, requires confirmation of passive state. Even if the subjects are asked to relax, spontaneous bursts of muscle activity are regularly found. Not including EMG measurement does not allow for elimination of trials with activity and is thus another factor that could have contributed to the large variability in the data set.
- MVC were assessed without controlling for the level of activation in the participants. As not all people can activate their muscles maximally, this will be another source of variability. However, this limitation was not mentioned in the paper.
- For the estimation of force exerted by the quadriceps muscles, several assumptions and estimates were used. Instead of individual assessment of moment arms (see for example papers by Maganaris), data from the literature was used. Furthermore, distribution of forces according to the PCSA based on the literature was calculated (see also point about PCSA above), while large interindividual differences are expected.
Together with the limitations acknowledged, it will be very hard to confirm the theoretical relationships between muscle architecture and mechanics using in vivo measurements.

The authors hypothesize a relationship between fascicle length and force at high shortening velocities. Especially at low forces and for muscles with large pennation angles, it has been shown that fiber rotations (e.g. changes in pennation angle) play a very important role (see work by Azizi and Roberts on muscle gearing). With the data collected, the role of this parameter for force at high shortening velocities could be assessed.

See additional comments for specific comments.

Validity of the findings

See other comments

Additional comments

Specific comments
- ln 30; it is not described
- ln 39; the authors suggest that 60 degrees is close to optimum length, but they manipulate joint angle. Then it is more consistent to describe the conditions imposed as such.
- ln 40; the conclusion does not contain much information. Which limitations? Which challenges?
- ln 75; the authors frequently jump between levels of organization (from sarcomere to fiber and fascicle). In most cases, in vivo studies do not measure phenomena at the sarcomere and fiber level and it is thus more appropriate to talk about fascicles.
- ln 84; How is a fascicle length of the combined quadriceps assessed?
- ln 102; not clear what the authors mean to indicate with ‘synergistic force’
- ln 133; not clear if this is before or after the warming up described below.
- ln 163; fascicle length with the cosine of pennation angle will provide the component of the fascicle contributing to muscle length, but based on one fascicle length muscle length cannot be assessed. What else was used to do this?
- ln 168; what exactly was the ankle angle used for the ‘standardized position’?
- ln 178; How many distances were included in the mean to calculate muscle thickness?
- ln 184; how was the probe holder attached to the leg?
- ln 187; please indicate along which axis the angle was manipulated. Also who the images from two angles were combined for image analysis. Was the angle set at 5 degrees or assessed for each subject?
- ln 205; the orientation of the upper aponeurosis is not the same as the insertion of the superficial fascicle.
- ln 222; not clear where the coordinates are coming from.
- ln 235; I suggest to change force-sok with force-so400.
- ln 244; Did the authors consider performing a multiple regression analysis? Why not?
- ln 267; At which joint angle was the optimal fascicle length found? This would be providing evidence for the statement that this should be around 60 degrees. As indicated above, this would also have been an opportunity to test each subject at the same position on their length-force curve.
- ln 303; please replace ‘linked’ with an appropriate word
- ln 325; ‘regional differences’ of which variable?
- ln 358; what about muscle gearing and effects on shortening velocity?
- ln 364; not clear why this is surprising. As CSA and PSCA can also be assessed, why then would one chose for a less informative parameter?
- ln 389; Here the authors assume that behavior assessed at the myofibril level can be extrapolated to the in vivo level. There are observations from whole limbs that suggest distribution of sarcomere length.
- ln 391-392; This sentence is unclear and should be reworded.
- ln 397; muscle slack length could have been assessed using the US methods and hence eliminated as a confounding factor (see above).
- ln 399-400; Sentence is unclear.
- ln 402 ‘shear strains from other muscles’ – what do the authors mean here?
- ln 405; what was the size of the artefact? Or how much were the parameters affected by it?
- ln 436; please indicate for what the VL was representative?
- ln 446; ‘direct relationship’ – not clear how this differs from an indirect relationship.
- Figure 4; not clear why Fmax was not 1 for all subjects.
- Table 1; units for pennation angle and thickness are missing.

Reviewer 2 ·

Basic reporting

I read with interest and pleasure this manuscript. This is a needed study in my opinion, as it addresses very important issues related to muscle architecture analysis in different muscle conditions.
I have few comments that are mostly methodological: even if the authors have taken care of many details regarding the measurements of architecture, the choice of use of the modified SMA algorithm could have lead, in this case, and in my humble opinion, to inaccuracies. Also, the fact that the double probe method was employed and that, unfortunately, the proximal scans couldn't be used to account for curvature of the fascicles (especially when this becomes important at different joint angles), could have influenced the association with the functional parameters. Nevertheless, I applaud the authors for a good study design and an enjoyable manuscript.

Experimental design

The experimental design is sound and precise.
It is indeed a shame that the double probe method did not lead to the use of the proximal scans and therefore they couldn't be used to account for the curvature of the fascicles.

Validity of the findings

The validity of the findings is high and of relevance for the muscle physiology and mechanics fields

Additional comments

1- Because VL was the only muscle investigated (and I understand why, methodologically and physiologically speaking), don't the authors think that the associations presented fall a bit short? This has been mildly addressed in the limitations section, but I think this should be stressed a bit more. And maybe the authors could stress that often researchers use VL as a surrogate of the whole quadriceps function (many studies draw conclusions on the basis of data obtained by one muscle), but that this maybe is not completely correct when speaking about architecture and function (especially based on your results?)?

2- line 72 - Maybe the authors could briefly explain the results of Wakahara 2013 and also reporting what was the explanation given for the lack of associations between PA and function?

3- Lines 94-95 - Can the authors explain a bit more why the also employed panoramic scans? Maybe reporting studies that compared panoramic vs. b-mode static US scans?

4- Line 127 - was this 3Dprinter pad adapted to each subject? if not, what does it add to the test compared to the manufacturer's pad?

5- Figure 1 - This may represent a methodological problem -> using an automated algorithm is obviously a good choice, as it objectifies the method. But what if a scan was collected in a point where architecture is not homogeneous? VL architecture becomes different from its mid belly one around 35% of femur length (and in fig 1 it seems that the low left end of the picture is presenting a different architecture from the right end). If SMA is employed, then the algorithm averages the orientation of ALL fascicles that are visible in a given scan, thus averaging two potentially different architectural arrangement within the muscle.. Could this lead to values that are a description of an architectural arrangement that is not representative to the real one? And could these differences be more emphasised during contraction? How did the authors make sure that only the fascicles belonging to the mid-muscle portion were averaged/evaluated/digitised?

6- Figure 2 represents the other potential methodological issue (IMO) - When using the proximal scan to check the aponeurosis inclination, how were you sure that the inclination was right?
Moreover, it is a shame that you could not account for curvature here. This could have influenced your results. Maybe this is something to stress in the limitations, but maybe stating that this is something that you could take care of the future studies?

7- Line 282 - please check the text - there is an error/mistake (maybe a refuse from previous versions). (Or maybe it's my PDF version that is showing something wrong).

8- Line 311 - Lichtwark 2018 is a perfect reference for this, you could also consider citing Pincheira et al. 2022 to support your statement. They were data on a different muscle than TA (BFlh) that showed regional differences at baseline and after 3 weeks of resistance training.

I like the whole discussion, especially the part related to the missing correlation to MVC60 due to probable bulging or shear strains from other muscles. I would stress even more that what we measure in a static situation may not be related to the behaviour of fascicles and muscle shape during contraction, although we would like this to be!

---

## Round 0.2 · accepted · Accept

The authors have adequately addressed the reviewers' concerns. This paper makes a valuable and critical contribution to the literature on muscle mechanics.

·

Basic reporting

All comments were addressed adequately.

Experimental design

All comments were addressed adequately.

Validity of the findings

All comments were addressed adequately.

Additional comments

All comments were addressed adequately.

Reviewer 2 ·

Basic reporting

The authors have addressed all my comments.

Experimental design

The experimental design is sound and precise.
My doubts on the double probe array methods have been addressed by the authors' rebuttal

Validity of the findings

The validity of the findings is high and of relevance for the muscle physiology and
mechanics fields

Additional comments

The authors have addressed all my comments.
Thank you for having considered all my points in the revised manuscript